# PDKit: A data science toolkit for the digital assessment of Parkinson's Disease

**Cosmin Stamate**, **Joan Saez Pons**, **David Weston**, **George Roussos** *

Department of Computer Science and Information Systems, Birkbeck College, University of London, London, United Kingdom

* g.roussos@bbk.ac.uk

**Data Availability Statement:** All relevant data are within the manuscript and its Supporting information files. All relevant files are available from https://github.com/pdkit/pdkit.

## Abstract

PDkit is an open source software toolkit supporting the collaborative development of novel methods of digital assessment for Parkinson's Disease, using symptom measurements captured continuously by wearables (passive monitoring) or by high-use-frequency smartphone apps (active monitoring). The goal of the toolkit is to help address the current lack of algorithmic and model transparency in this area by facilitating open sharing of standardised methods that allow the comparison of results across multiple centres and hardware variations. PDkit adopts the information-processing pipeline abstraction incorporating stages for data ingestion, quality of information augmentation, feature extraction, biomarker estimation and finally, scoring using standard clinical scales. Additionally, a dataflow programming framework is provided to support high performance computations. The practical use of PDkit is demonstrated in the context of the CUSSP clinical trial in the UK. The toolkit is implemented in the python programming language, the de facto standard for modern data science applications, and is widely available under the MIT license.

## Author summary

Parkinson's Disease is the fastest growing neurological condition affecting millions of people across the world. People with Parkinson's suffer from a variety of symptoms that result in diminished ability to move, eat, remember or sleep. Research in new treatments are limited because the clinical tools used to assess its symptoms are subjective, require considerable time to perform and specialised skills and can only detect coarse-grain changes. To address this situation, clinicians are turning to smartphone apps and wearables to create new ways to assess symptoms that are more sensitive to change and can be applied frequently at home by patients and their carers. In this paper, we discuss PDkit, an open source toolkit that we developed to help address this current lack of algorithmic and model transparency. Adopting PDkit facilitates the open sharing of standardised methods and can accelerate the development of new methods and system to assess Parkinson's and enables research groups to innovate. The toolkit provides funcionality that support data ingestion, quality of information augmentation, feature extraction, biomarker estimation and finally, scoring using standard clinical scales. The practical use of PDkit is demonstrated via its use by the CUSSP clinical trial conducted in the UK.

**Funding:** GR, DW, CS and JSP acknowledge support by the Michael J. Fox Foundation for Parkinson's Research (MJFF) with Grant ID 14781 to GR & DW for the project entitled "A Scalable Computational Data Science Toolbox for High-Frequency Assessment of PD" awarded under its Computational Science 2017 programme. The funders had no role in study design, data collection and analysis, decision to publish, or preparation of the manuscript.

**Competing interests:** The authors have declared that no competing interests exist.

This is a *PLOS Computational Biology* Software paper.

## Introduction

Parkinson's is the second most common neurodegenerative disease across the globe with as many as 10 million patients worldwide. Parkinson's Disease (PD) is caused by the degeneration of dopaminergic neurones, a group of neurones located in the mid-brain which are the main source of dopamine in the human central nervous system [1]. They play a crucial role in the control of voluntary movement, mood, reward, addiction, stress and critically, the reward system that controls learning. Although the cause for the loss of dopaminergic neurones in PD is not yet known, their selective degeneration results in the distinctive presentation of Parkinson's commonly associated with a wide spectrum of motor and cognitive symptoms including tremor, slowness of movement and freezing, muscular stiffness, poor postural stability, sleep-related difficulties, depression and psychosis [2].

Since there is currently no cure, clinical care pathways for PD are focused on symptom management, a life-long process that typically includes pharmacological treatment, physiotherapy and, at the advanced stages of the disease, surgery [3]. A key ingredient of the pharmacological regime is medication with levodopa, prolonged use of which in most cases results in patients developing side effects such as dyskinesias [4]. Moreover, symptoms are highly heterogenous across and within patient profiles and PD progresses at different rates in different individuals. The golden standard for the assessment of symptoms is the application of established clinical measures by specialist practitioners, in particular the Universal Parkinson's Disease Rating Scale (MDS-UPDRS) [5]. However, the limited sensitivity of these methods restricts the effectiveness of motor symptom severity assessments. Data revealing individual symptom variability and observed trends in particular are scarce, further restricting opportunities to evaluate new treatments.

To address the challenges towards achieving sensitive, frequent and objective PD assessment, digital methods have been increasingly considered over the past decade as a promising complementary approach with distinct advantages. In particular, the wider availability of smartphones and wearables offers the promise to enable the unsupervised, and at high-frequency or continuous measurement of motor and non-motor performance across the global patient population [6–8]. Mirroring patterns of contemporary data production in other domains, digital assessment represents a paradigm shift in the clinical assessment of PD potentially leading to the tremendous increase in the availability of patient performance data. In this setting, manual analysis of data is no longer viable. Instead, it is imperative to adopt a software-based approach so that outputs of clinical relevance can be computed automatically and presented to researchers, clinicians and patients in an intuitive manner.

Nevertheless, despite the rapid proliferation of proposals for digital assessment methods, only limited progress has been made towards establishing robust and generalisable digital endpoints such that would match the expectations of medical regulators for use in clinical trials. To a certain extent, this is due to common challenges that other medical domains have faced early on in the development of digital assessments: Small study samples, feature selection bias and failure to replicate results due to differences in sensor placement and calibration, as well as lack of clarity in the use of analytical techniques, have led to a fragmented research landscape that hinders the development of effective techniques. PD studies have often failed to account for inter-rater variability leading to machine learning models also learning subjective bias. Moreover, the highly

heterogeneous presentation of PD cannot be fully captured by monitoring only a small subset of symptoms thus resulting in too blunt an instrument failing to capture critical symptoms and hence being unable to define a deep phenotype at the individual level. Last but not least, digital assessment studies are at high risk of providing over-optimistic results due to feature selection bias if a large number of post hoc candidate digital features or machine learning algorithms are tested within a limited size study resulting in highly unstable features that do not generalise.

To address the lack of algorithmic and model transparency in particular, we designed and developed PDkit, a comprehensive software toolkit for the management and processing of patient data captured continuously by wearables (passive monitoring) or by high-use-frequency smartphone apps (active monitoring). The toolkit facilitates the application of a data science methodology to the analysis of such data and in particular makes it possible to clearly describe the computational steps involved. Our motivation is to provide an open and inclusive framework that can be used to capture all steps of analytical processing in detail, thus providing a key ingredient towards clarity and reproducibility of findings. PDkit is implemented in the python programming language, the de facto standard for modern data science applications, and is made available as free/open source software under the MIT license, which permits all uses without restrictions.

In developing PDkit, we draw inspiration by the successes of similar initiatives in other areas of digital healthcare, which achieved breakthroughs by adopting an open approach such as the ADNI initiative in Alzheimer's for fMRI imaging [9] and the SPM software toolkit for the analysis of brain imaging data sequences [10] among others. Specifically, using PDkit as an enabler of a Digital Health Technology (DHT) processing pipeline in Parkinson's, has the following advantages:

- It supports the development and open sharing of standardised methods that allow the comparison of results from multiple centres and hardware variations.

- It provides access to advanced knowledge and domain expertise relating to software engineering, signal processing and machine learning algorithms. Hence, it reduces the costs of bespoke software development in DHT-based exploratory research and clinical studies, especially those related to software implementation and verification.

- It enhances confidence in the computational outcomes produced in studies, due to the fact that the software is tested by a large user community.

- It provides concise, domain-specific programming abstractions specifically targeting Parkinson's, thus eliminating the need to write repetitive code.

- Contrary to proprietary software, it provides the ability to inspect the algorithms employed and their implementation, thus facilitating the in-depth exploration of the results generated and of any clinical inferences made.

By releasing PDkit as open source software, our long-term goal is to support therapeutic development and cost-effective clinical trial evidence collection. In particular, we endeavour to facilitate the explicit definition of clinical outcome measures and to help identify the advantages and limitations of specific quantitative metrics of disease progression so that the research community can converge on effective generalisable digital digital endpoints suitable for clinical trials.

## Design and implementation

PDkit has two core ingredients: first, it adopts the information-processing pipeline abstraction, a well-established data science design pattern, specifically tailored to the assessment of

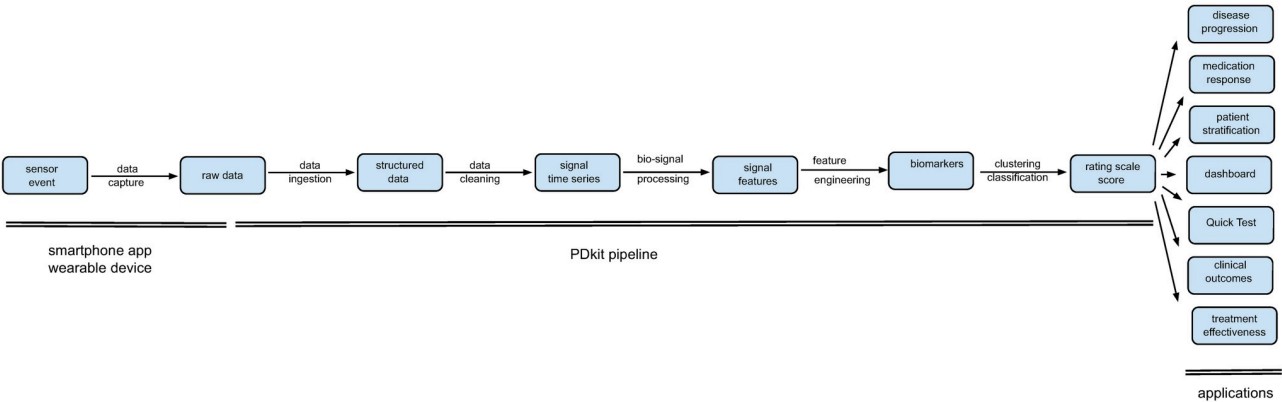

**Fig 1. PDkit pipepline.** PDkit information-processing pipeline for Parksinon's.

Parkinson's (as depicted in Fig 1). PDkit pipelines start with a data ingestion stage designed to consume wearable and smartphone app measurements in a wide variety of formats, followed by quality of information tests, feature extraction, biomarker estimation and finally scoring using standard clinical scales such as MDS-UPDRS. Each pipeline can be terminated at any intermediate processing stage as appropriate for a particular investigation. Each stage, typically implemented as a distinct python class, provides information export and import so that intermediated results can be extracted easily and so as to permit the execution of the pipeline in a staggered manner.

The second ingredient in the design of PDkit is support for two alternative programming models: a standard python programming interface and also an alternative dataflow programming model supported seamlessly via the core API specification. The motivation for adopting this approach is our desire to balance the need for a low barrier of entry for developers so as to encourage the adoption of this toolkit; and at the same time, to cater to modern scalable information processing architectures, which are critical in deploying digital assessments at population scale. Moreover, the combination of an information pipeline approach and an adaptable programming model enables PDkit to effectively support both active and passive monitoring within an integrated framework.

## Data ingestion

Considering each stage of the PDkit pipeline sequentially (from left to right), the data ingestion step supports a range of sensor modalities and data formats. Active monitoring, typically implemented as a smartphone app, is carried out in distinct measurement sessions resulting in a data file set containing timed sensor measurements. There is currently no widely adopted standard for the encoding of such data, hence each app adopts a custom approach: the mPower app [7] for example uses a proprietary JSON schema while the cloudUPDRS [8] and Hopkins PD [11] apps employ a simple flat CSV-formatted (comma separated values) text file. For the current list of supported apps please refer to the online documentation of PDkit available on Read-the-Docs [12]. Passive monitoring using wearables typically employs a gateway device, in most cases a smart home hub such as the Apple HomePod and Amazon Echo, or a smartphone for data streaming over a low-power wireless interface (typically Bluetooth Low Energy). The gateway subsequently employs one of several streaming protocols to relay the measurements for further processing in near real-time. To this end, popular streaming

approaches include MQTT [13] and publish-subscribe schemes [14] with both approaches currently supported by PDkit, the latter via the Google Pub-Sub API [15].

Regardless of the data input mode (batch or streaming) and format, ingested raw data is converted and stored internally in standardised symptom-specific PDkit representation. Symptom-specific time series representations are derived from Pandas [16], a popular specification for python-based data science applications, and include reaction, gait, tremor, and tapping measurements incorporated in python classes such as `TremorTimeSeries` and `FingerTappingTimeSeries` for tremor and finger tapping measurements correspondingly. The exception to this are voice samples, which are treated as binary large objects. PDkit is designed to be inherently extensible so that connectors to additional data files and streaming formats can be easily added as required.

## Quality of information

Having converted raw sensor measurements to PDkit native representations, the next processing stage is to assess, and when necessary improve, the quality of the data recorded. Tests relate to typical data integrity checks such missing and out of range values or other outliers caused by transmission errors or sensor malfunction, consistent indexing and standardised labelling. A second group of QoI transformations relate to time series resampling to normalise fluctuations in measurement regularity, which for example would hinder the application of Fast Fourier Transforms (FFT), and downsampling for data reduction and improved manageability. Finally, a third group of processing functions relate to improvements in the relevance of observations such as bilateral truncation of time series to account for start up and cool down effects in test performance and verification of movement correctness in the case of active monitoring to validate unsupervised data collection [17] as well as the injection of higher-order quality features such as data augmentation and signal segmentation [18].

## Feature extraction

The next stage in the pipeline involves the extraction of distinctive data features for each symptom datatype using processor classes such as `VoiceProcessor` and `GaitProcessor`. For a typical active monitoring session, PDkit can calculate over 800 different features reflecting the plethora of signal processing and machine learning techniques suggested in the literature. There are two schools of thought regarding methods to generate and select features: One approach suggests that features employed for symptom assessment should reflect biomedical intuition based on clinical experience as well as what is important to patients, with the opposing view exhorting the advantages of a purely data-driven approach. PDkit caters to both viewpoints for example providing implementations of most standard bio-inspired features found in the literature for PD. To further illustrate this approach, note that a popular feature used to characterise kinetic tremor and tremor at-rest is calculated as the cumulative magnitude of the scalar sum acceleration across three axes for all frequencies between 2 Hz and 10 Hz. To calculate this metric, PDkit obtains the tremor power spectrum by first filtering the tremor time series with a Butterworth high-pass second order filter at 2 Hz and subsequently applying an FFT to the filtered waveform data. Similarly, the assessment of the pronation-supination movements and leg agility tests, requires the estimation of the frequency and power of movement. To obtain these, the PDkit first removes the DC offset and then applies a Butterworth low-pass second order filter at 4 Hz to exclude most of the tremor. Finally, the power of movement is calculated as the total amplitude between 0 and 4 Hz and the frequency derived from the power spectrum.

PDkit caters to the data-driven approach through the implementation of a wide range of time series and voice characteristics that are commonly used in analyses of similar data but are not necessarily associated to clinical observations. Several of these calculations are inherited by incorporating third party python modules notably, TSFRESH for time series [19] and Praat [20] for voice analysis. Similar to all PDkit elements, feature extraction is implemented in an extensible manner so that additional feature extraction techniques can be easily incorporated.

## Digital biomarkers

A key requirement for predictive analytics for disease progression and patient stratification is the identification of digital biomarkers with strong inferential properties, that is, indicators that match higher-level clinical insights obtained from the combination of carefully selected lower-level signal characteristics captured by sensors. To this end, PDkit supports two distinct types of biomarkers: standard digital biomarkers as typically encountered in the literature, which correspond to a unitary (in time) set of measurements of symptoms, and typically expressed in the form of a feature vector as typically employed in data science within a standard feature engineering approach. The second, and arguably more interesting type of biomarker, are so-called longitudinal biomarkers that result from the accumulation of features extracted from repeated measurement of symptoms over an extended period of time. For example, a longitudinal digital biomarker can be constructed by calculating the descriptive statistics of the distribution of the same feature (such as power of the dominant tremor frequency) using measurements of the same symptom (such as kinetic tremor of the left hand) sampled at various times during a week-long monitoring session. Such longitudinal biomarkers are of increasing interest for the effective assessment of PD [21] due to increasing evidence of their ability to cope with heterogenous disease presentation typical in PD. In this case, instead of focusing on individual snapshots in time, longitudinal biomarkers capture the aggregate characteristics of their statistical distribution which appear to provide a more consistent and sensitive way to characterise disease presentation.

## Clinical scores

The final processing stage in the PDkit pipeline involves training a predictive model of scoring with a clinical rating scale. In keeping with the inclusive approach adopted, PDkit again supports two alternatives: first, a data-driven methodology employing a repository of patient data and suitable clustering algorithms is used to discover mappings between biomarkers and their corresponding grades in the rating scale. Alternatively, when clinical labelled data are available, a supervised machine learning approach can be adopted using class `ClinicalUPDRS` that employs a family of classifiers to generate rating scale level inferences. In either case, at the end of this processing stage it becomes possible to receive new passive or active sensor measurements and convert them fully automatically to a clinical MDS-UDPRS score without the involvement of a human rater. As a consequence, a PDkit-based model can be employed for the end-to-end automatic assessment of patients thus leading to a variety of applications that depend in such automation for example monitoring disease progression, tracking responses to medications and treatments, and patient stratification. Currently, only MDS-UPDRS grading is supported as this is the most common rating scale in use and the only recognised by the FDA and EMA for clinical studies, however other rating scales can be easily accommodated due to the extensible design of the toolkit. It is envisioned that PDKit can be used not just for the development of effective biomarkers but also for clinical outcome assessments in that there is potential to replace the MDS-UPDRS clinician-reported outcome.

### Implementation and release

PDkit was developed with support by the M.J. Fox Foundation for Parkinson's Research under their computational science programme. Throughout its development it has been available as open source under the MIT License, with the first official release v1.0 becoming available in October 2018. At the time of writing the current release is v1.3.2 incorporating additional support for voice assessments and Hopkins PD/OPDL result datasets. The dataflow programming model is implemented using Apache Beam [22]. The methods implemented within PDKit have been carefully curated and where relevant the citations are provided.

Since the release of version 1.0, the toolkit has been downloaded over 75, 000 times through the PyPI repositories, which allow easy installation of python language modules. Although PyPI downloads are inherently anonymous and thus not possible to identify specific uses and user groups, the development team is independently aware that PDkit is actively used in Parkinson's clinical studies by universities and commercial companies in Belgium (Leuven), Germany (Meinheim), France (Grenoble), Italy (Milan), U.K. (London) and the USA (Miami).

## Results

As noted earlier, a key motivation in the development of PDkit is the recognition that digital assessments of motor severity could significantly improve the sensitivity of clinical trials and personalise treatment in PD but face considerable challenges before they can be widely adopted. The ability of digital biomarkers in particular to capture individual change across the heterogeneous motor presentations typical of PD, remains inadequately explored against current gold-standard clinical reference standards. To determine the validity and accuracy of subject-level smartphone-based measures of severity in PD across a number of motor subitems, in October 2016 we initiated The CloudUPDRS Smartphone Software in Parkinson's Study (CUSSP) in collaboration with the UCL Institute of Neurology and the outpatient departments of the National Hospital for Neurology and Neurosurgery and Homerton University Hospital in the UK [23].

CUSSP was a prospective, dual-site, crossover-randomised study comparing structured single-time-point smartphone-based and multiple blinded clinical rater assessments of motor severity with recruitment completed in May 2019. In total, sixty adults were enrolled with early to mid-stage idiopathic PD without dementia. The study protocol included the video-taped administration of the 33-item Part III of MDS-UPDRS and a 16-item smartphone-based assessment using the cloudUPDRS app [8] in randomised order. The primary outcome was the degree to which subject-level smartphone-based measures calculated using a study-specific PDkit pipeline (the index test) predicted subject-level Part III MDS-UPDRS subitems as assessed by three blinded clinical raters (the reference-standard). This was quantified as the leave-one-subject-out cross-validation (LOSO-CV) predictive accuracy of a range of features and machine learning algorithms implemented using PDkit.

### CUSSP data analysis using PDKit

In order to demonstrate how a practitioner might use PDKit, we briefly sketch how the statistical analysis of the raw data obtained from the smartphone app was performed within CUSSP. We focus on one subitem, tremor, specifically the kinetic tremor of the left hand. The analyses presented below can be found in the Jupyter Notebooks entitled `01—TremorProcessor` and `05—ClinicalUPDRS` and sample data files are provided in the `tests/data` directory of the source code distribution.

**Initialisation and data ingestion.** The raw data comprises timestamped 3D accelerometer readings recorded at the maximum sampling rate supported by each smartphone used (50 Hz

in the case of the sample file below) and is stored in comma separated format recorded by the cloudUPDRS app. This data is ingested using the `load` method from the `TremorTime-Series` class. Note that the APIs provided by PDKit follow an object-oriented approach in order to clearly organise the diverse set of available functionality. This method returns the data in a Pandas dataframe. Intermediate data structures used in PDKit are represented using standard Pandas dataframe or timeseries, this ensures that practitioners are not 'locked-in' to using only the methods available from within PDKit and are able to conveniently incorporate their own extensions. In addition displaying results can be achieved simply by using standard graphics libraries such as matplotlib and seaborn directly on these data structures. This is useful for a wide variety of tasks including exploratory analysis and sanity checking.

```
import pdkit
filename = 'T-KINETIC_TREMOR_OF_HANDS-LEFT_HAND-_2458.csv'
ts = pdkit.TremorTimeSeries().load(filename)
```

**Feature extraction.** As described earlier, one relevant feature for tremor is the cumulative magnitude of the scalar sum acceleration across three axes over a specific range of frequencies. This value is computed using methods from the `TremorProcessor` class as follows: resample the original signal using to ensure uniform spacing of samples in the time domain, apply a high pass filter using, convert data to the frequency domain, and, finally we calculate the specific feature:

```
tp = pdkit.TremorProcessor()
resampled_data_frame = tp.resample_signal(ts)
filtered_data_frame = tp.filter_signal(resampled_data_frame)
fft_data_frame = tp.fft_signal(filtered_data_frame)
amplitude, frequency = tp.amplitude_by_fft(fft_data_frame)
```

To promote economy of expression and clarity, the above steps can alternatively be carried out in a single step using PDkit as they represent a common processing pipeline employed extensively in the literature:

```
tp = pdkit.TremorProcessor()
amplitude, frequency = tp.amplitude_by_fft(ts)
```

**Clinical scoring.** Aggregate results computed through the above feature extraction process (in the example below summarised in a spreadsheet) can be used to build a classifier to predict MDS-UPDRS Part III subitem ratings. When the test data are labelled by an experienced clinician, supervised classification can be employed aiming to match the performance of a human rater:

```
clinical_UPDRS = pdkit.Clinical_UPDRS(labels = 'updrs_scores.
csv', features)
clinical_UPDRS.predict(measurement)
```

To train the classifier two parameters are required: a dataframe of features and the name of the file that contains the corresponding clinical assessment. Predictions are carried out using the `predict` method. The typical performance evaluation scenario which requires that leave-one-subject-out cross-validation is carried out, can be achieved by combining the code above and the well known python package `sklearn`.

## CUSSP results

Data analysed included 990 smartphone tests and 2, 628 blinded Part III MDS-UPDRS subitem ratings. A fully pre-specified LOSO-CV analysis averaged over all 16 subtests classified 70.3%– with Standard Error of Mean (SEM) 5.9% of subjects into a similar category to a blinded clinical rater. This outperformed constant (58.0%; SEM 7.6%) and random (36.7%;

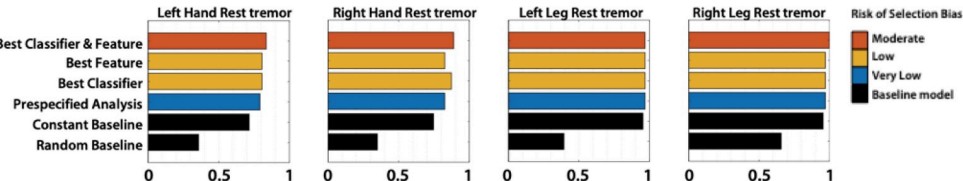

**Fig 2. CUSSP study results of PDkit predictive performance.** Item-level performance based on LOSO-CV calculations for four tremor tests conducted using the cloudUPDRS app.

SEM 4.3%) baseline models. Post hoc optimisation of PDkit classifier and feature selection improved performance further to 78.7% (SEM 5.1%). Individual subtests had a variable LOSO-CV accuracy ranging between 53.2%–97.0% due to variation in category balance across subtests and variable classifier learning. These results strongly suggest that smartphone-based measures of motor severity have predictive value for clinical measures at the subject-level with substantial variability observed depending on the body part or clinical feature tested. An illustration of item-level performance can be seen in Fig 2 and full study details in [23].

## Availability and future directions

PDkit is available as open source software on GitHub [24] as well as a packaged python module via the Python Package Index (PyPI) repository [25] to facilitate easy installation for most users (cf. S1 Compressed ZIP File. To support uptake, we have published extensive documentation on Read-the-Docs [12] as well as a comprehensive collection of Jupyter Notebooks demonstrating the key elements of functionality of the toolkit. Notebooks are intended to demonstrate key use case studies of the toolkit using sample data sets from the cloudUPDRS [8] and mPower [7] projects (sample data also released on the PDkit GitHub repository). Sample analyses can be executed both natively or via popular notebook hosting sites for collaborative research such as Google's Collaboratory (cf. colab.research.google.com) to further depress the initial learning curve. Additional data can be obtained by interested researchers under license via the Sage Bionetworks Digital Health Data Repository and the OPDC Project at the University of Oxford [6]. These datasets are fully supported by PDkit and thus can be ingested directly using the notebooks.

Going forward, the PDkit development team aims to continue to provide support to all groups and individuals expressing an interest to explore the use of the toolkit or to contribute additional or novel methods and techniques. Of particular interest is to support further smartphone or wearable-based studies designed to mitigate against subjective and feature selection bias and assessing performance across a range of motor presentations to avoid overly optimistic performance estimates. In this regard, a particular area where PDkit can make a significant contribution is to increase the transparency of algorithms employed for the extract of digital biomarkers. This is also a key priority identified by the Critical Path for Parkinson's initiative [26] and PDkit has engaged in collaborative work to help reach this goal.

## Conclusion

In this paper, we provided an overview of PDkit, a python-based open source toolkit for the digital assessment of symptoms of Parkinson's Disease. The use of PDKit in the analysis of CUSSP made it straightforward to perform not just the analysis based on pre-specified features that used a standard statistical classifier, but to also subsequently perform a very broad exploratory analysis over a large number of features and classifiers. Furthermore, we believe that

PDkit provides a key ingredient towards enhancing algorithmic transparency of digital assessments for PD through open sharing of analytical methodologies and their concrete implementation as software artefacts. This open and inclusive process can help establish the tradeoffs involved in alternative proposals and thus build consensus on which candidate methods can deliver effective mobile and wearable digital assessments for clinical use.

## Supporting information

**S1 Compressed ZIP File. PDkit source archive.** Exported from current GitHub repository including source code, data and documentation.
(ZIP)

## Author Contributions

**Conceptualization:** David Weston, George Roussos.

**Funding acquisition:** David Weston, George Roussos.

**Investigation:** Cosmin Stamate, Joan Saez Pons, David Weston, George Roussos.

**Methodology:** David Weston, George Roussos.

**Project administration:** David Weston, George Roussos.

**Resources:** Cosmin Stamate, Joan Saez Pons.

**Software:** Cosmin Stamate, Joan Saez Pons, David Weston, George Roussos.

**Supervision:** David Weston, George Roussos.

**Validation:** Cosmin Stamate, Joan Saez Pons, David Weston, George Roussos.

**Writing – original draft:** David Weston, George Roussos.

**Writing – review & editing:** Cosmin Stamate, Joan Saez Pons.

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
