## [Decision Letter · Decision Letter 0]

24 Sep 2020

Dear Prof. Dr. Roussos,

Thank you very much for submitting your manuscript "PDKit: A data science toolkit for the digital assessment of Parkinson’s Disease" for consideration at PLOS Computational Biology.

As with all papers reviewed by the journal, your manuscript was reviewed by members of the editorial board and by several independent reviewers. In light of the reviews (below this email), we would like to invite the resubmission of a significantly-revised version that takes into account the reviewers' comments.

We cannot make any decision about publication until we have seen the revised manuscript and your response to the reviewers' comments. Your revised manuscript is also likely to be sent to reviewers for further evaluation.

Sincerely,

Dina Schneidman

Software Editor

PLOS Computational Biology

Reviewer's Responses to Questions

**Comments to the Authors:**

Reviewer #1: The paper presents PDKit, a software tool aimed at collecting and analysing data related to Parkinson’s Disease and derived from wearables and behavioural monitoring. The last step of the tool is to provide a prediction of the severity of the disease with respect to standard clinical evaluations.

The paper can be classified as a technical paper, describing the main software modules composing the tool and the general features of each module, presenting only some simple application examples.

The presented results derive from previous studies on big datasets collected during experimental campaigns. These results have been already published by the authors in dedicated papers.

The main novelty of the paper is the presentation of the software architecture of the tool, which is also available to researchers and it has been already dowloaded several times. However, the description of the tool lacks of specific examples of its applications, for example referring to the experimental results presented in Section “Results”. The authors claim that they provide specific examples and use cases in notebooks, available for download, and in the Read-the-Docs files, but not describing in detail this part in the paper, it reduces the novel contribution of the paper.

In addition, a more detailed description of the advantages of using the tool, for example from the medical users (in terms of usability and acceptance) would provide an important added-value for the paper. For all these reasons, I suggest the authors to review the paper including these missing parts.

Reviewer #2: This manuscript describes a tool kit developed by the authors with “to facilitate the development and open sharing of novel digital biomarkers for PD and hence help address the current lack of algorithmic and

model transparency ».

This goal is extremely worthwhile for PD and other diseases, where digital biomarkers and digital clinical outcome assessment are becoming increasingly widely used, often without well characterised performance or transparent algorithms. And the point is well made that this is a barrier to acceptance of these methods by regulators.

The content is very interesting, but I think readers may find it hard to follow the flow of the paper, as the results section that involves results obtained by processing data from CUSSP, seems out of place, without proper method or discussion of the results. These results are from quite a large data volume, and I was surprised to see so little comment on them and no reference in the conclusions.

I would suggest a refinement of the structure to address this concern, including expanding these results, having a clear description of how the toolkit enabled this analysis, plus discussion ad reference in conclusions, so that readers can get an example of the application of the tookit to generate novel results.

More minor comments.

Line 139: “One approach suggests that features employed for symptom assessment should reflect biomedical intuition based on clinical experience, with the opposing view exhorting the advantages of a purely

data-driven approach”. A patient rather than clinical experience perspective should also be mentioned her, see FDA patient focused drug development programmes.

Line 162. Given the focus on regulatory issues, please clarify some of the terminology, and in particular the difference between Biomarker and Clinical outcome assessments (see various references on the EMA and FDA web site https://www.fda.gov/drugs/development-approval-process-drugs/drug-development-tool-ddt-qualification-programs). The digital technology that are the focus of this paper might be either biomarkers or clinical outcome assessments, but more likely the latter.

**Have all data underlying the figures and results presented in the manuscript been provided?**

Reviewer #1: Yes

Reviewer #2: Yes

PLOS authors have the option to publish the peer review history of their article (what does this mean?). If published, this will include your full peer review and any attached files.

Reviewer #1: No

Reviewer #2: **Yes: **Derek Hill
---

## [Decision Letter · Decision Letter 1]

24 Feb 2021

Dear Prof. Dr. Roussos,

We are pleased to inform you that your manuscript 'PDKit: A data science toolkit for the digital assessment of Parkinson’s Disease' has been provisionally accepted for publication in PLOS Computational Biology.

Best regards,

Dina Schneidman

Software Editor

PLOS Computational Biology

Reviewer's Responses to Questions

**Comments to the Authors:**

Reviewer #1: The authors addressed all the reviewers' comments providing detailed motivations. The paper is ready for publication.

**Have all data underlying the figures and results presented in the manuscript been provided?**

Reviewer #1: Yes

PLOS authors have the option to publish the peer review history of their article (what does this mean?). If published, this will include your full peer review and any attached files.

Reviewer #1: No

---

## [Editor Report · Acceptance letter]

9 Mar 2021

PCOMPBIOL-D-20-00762R1 

PDKit: A data science toolkit for the digital assessment of Parkinson’s Disease

Dear Dr Roussos,

I am pleased to inform you that your manuscript has been formally accepted for publication in PLOS Computational Biology. Your manuscript is now with our production department and you will be notified of the publication date in due course.

With kind regards,

Alice Ellingham
